# The Feasibility of Longitudinal Upper Extremity Motor Function Assessment Using EEG

**DOI:** 10.3390/s20195487

**Published:** 2020-09-25

**Authors:** Xin Zhang, Ryan D’Arcy, Long Chen, Minpeng Xu, Dong Ming, Carlo Menon

**Affiliations:** 1Department of Biomedical Engineering, College of Precision Instruments and Optoelectronics Engineering, Tianjin University, Tianjin 300072, China; xin_zhang_bme@tju.edu.cn (X.Z.); xmp52637@tju.edu.cn (M.X.); 2Menrva Research Group, Schools of Mechatronic Systems Engineering and Engineering Science, Simon Fraser University, Metro Vancouver, BC V5A 1S6, Canada; cmenon@sfu.ca; 3Schools of Engineering Science and Computer Science, Simon Fraser University, Burnaby, BC V5A 1S6, Canada; rdarcy@sfu.ca; 4Academy of Medical Engineering and Translational Medicine, Tianjin University, Tianjin 300072, China; cagor@tju.edu.cn; 5Tianjin International Joint Research Center for Neural Engineering, Tianjin 300072, China

**Keywords:** EEG, motor function, neural networks

## Abstract

Motor function assessment is crucial in quantifying motor recovery following stroke. In the rehabilitation field, motor function is usually assessed using questionnaire-based assessments, which are not completely objective and require prior training for the examiners. Some research groups have reported that electroencephalography (EEG) data have the potential to be a good indicator of motor function. However, those motor function scores based on EEG data were not evaluated in a longitudinal paradigm. The ability of the motor function scores from EEG data to track the motor function changes in long-term clinical applications is still unclear. In order to investigate the feasibility of using EEG to score motor function in a longitudinal paradigm, a convolutional neural network (CNN) EEG model and a residual neural network (ResNet) EEG model were previously generated to translate EEG data into motor function scores. To validate applications in monitoring rehabilitation following stroke, the pre-established models were evaluated using an initial small sample of individuals in an active 14-week rehabilitation program. Longitudinal performances of CNN and ResNet were evaluated through comparison with standard Fugl–Meyer Assessment (FMA) scores of upper extremity collected in the assessment sessions. The results showed good accuracy and robustness with both proposed networks (average difference: 1.22 points for CNN, 1.03 points for ResNet), providing preliminary evidence for the proposed method in objective evaluation of motor function of upper extremity in long-term clinical applications.

## 1. Introduction

Motor function assessment is crucial in the rehabilitation process following stroke, as it quantifies motor recovery and help the healthcare professionals determine the process of the rehabilitation training. Many questionnaire-based motor function assessments have been designed, such as the Fugl–Meyer Assessment (FMA) [1], National Institutes of Health Stroke Scale (NIHSS) [2] and Functional Independent Measure (FIM) [3], which require the participants to perform standard movements and get scores based on the examiners’ observation. Therefore, those assessments require prior professional experience to be administered properly and are not completely objective. It was reported that quantitative features calculated from event-related synchronization/desynchronization (ERD/ERS) correlates with motor function performance [4,5]. Anastasi et al. found that brain symmetry index (BSI) could be used stroke related assessments [6]. Leon-Carrion et al. suggested delta-alpha ratio (DAR) correlates with level of recovery in populations with acquired brain injuries [7]. Kawano et al., reported the large-scale phase synchrony (LPS) from rest electroencephalography (EEG) data correlate with FIM score [8], which could be a biomarker of post-stroke motor impairment and recovery [9]. Zhang et al. used EEG data from clicking a mouse to train an artificial neural network, which is able to quantify upper extremity motor function with very high accuracy (R > 0.9) [10]. Riahi et al. proposed a method using partial least squares correlation to identify the connectivity channels and frequency components of rest EEG data for FMA scores of upper extremity [11]. As discussed in the papers above, the feasibility of assessing motor function based on EEG data was investigated with one-time EEG recording. However, EEG is generally considered as a typical non-stationary signal [12,13]. Long-term applications based on EEG may suffer from its non-stationary nature and become unstable after weeks. None of the above studies investigate the performance of the proposed EEG-based motor function scoring methods in a longitudinal paradigm, which is extremely important in applications like monitoring recovery in stroke rehabilitation. The longitudinal accuracy and robustness of the above methods in scoring motor function is still unknown.

The aim of this paper is to investigate the feasibility of the EEG-based motor function assessment method in a longitudinal paradigm. As the convolutional neural network (CNN) method proposed in [10] showed highest linear correlation with the true FMA score. In addition, ResNet applies identity mapping to facilitate deep learning and minimize gradient vanishing, which suggested superior performance in other machine learning and pattern recognition applications [14]. Therefore, a deep residual neural network (ResNet) model was also trained and evaluated to explore further performance improvement by increasing network complexity. The two artificial neural network (ANN) models were trained with a pool data set collected in the previous study, which contains EEG data from 12 healthy participants and 14 participants with chronic stroke. These previously developed models were tested using EEG test data through a small sample validation study to evaluate the longitudinal performance of the ANN methods. Two case participants with chronic stroke, who also participated in the previous pool data collection, were evaluated using their data within and without inclusion in the model. The test data were collected within fourteen weeks in cooperation with a specific rehabilitative training process as described in [15]. In the longitudinal data collection program, three baseline assessments, two midterm assessments and one retention assessment were administered, with one participant involved in only the three baseline assessments. ANN model performances were evaluated as the absolute difference between the scores from EEG data and the FMA scores. The hypothesis predicted that both ANN methods would successfully track objective EEG changes of motor function and ability.

## 2. Materials and Methods

All the methods in this study were in compliance with the Declaration of Helsinki. The study was approved by the Simon Fraser University (SFU) Office of Research Ethics.

### 2.1. Inclusion Criteria

Participants were recovering from stroke and were >6 months post-injury. All participants were pre-screened based on age (35–85 years old) and cognitive statues (Montreal Cognitive Assessment (MoCA) greater than 23) [16]. Potential participants were excluded from this study if they: (1) had any neurological conditions in addition to the stroke impairment; (2) had unstable cardiovascular conditions or (3) had any other conditions that precluded them from following the protocol.

### 2.2. Participation Protocol

Two participants were recruited in this study. The study timeline is shown in Figure 1. Both participants first went through a one-month baseline assessment session consisting of three baseline assessments for motor function, once every two weeks. During the motor function assessments, FMA and Wolf Motor Function Test (WMFT) scores of upper extremity were administrated by an experienced “blind” examiner, who was not involved in the study development.

After the baseline assessments, one participant (P1) agreed to continue with the study following a rehabilitative training protocol designed by Orand et al. [15]. During the training period, the participant received the pre-defined training over three sessions per week, for six weeks (18 sessions in total). Clinical evaluation of motor function was assessed with FMA and WMFT on the 6th session and 18th session of the training period, which are referred as “Assessment 1” and “Assessment 2” respectively. An additional motor function assessment session was administered to the participant four weeks after the 18th session of the training period, which is referred as “Retention” session.

### 2.3. EEG Data Acquisition

In this paper, two data sets were used to investigate the longitudinal performance of the ANN models: the “pool data” and the “longitudinal data”. The pool data from [10] were used to train the ANN models in this paper. The pool data consists of 3640 trials collected from 12 healthy participants and 14 participants with chronic stroke. P1 and P2 also participated in pool data collection about 12 months before the longitudinal data collection. The longitudinal data were collected from P1 and P2 in 14 weeks and 4 weeks respectively, and were used to test the obtained ANN models in order to evaluate the longitudinal performance of the proposed motor function assessment method based on EEG. P1′s longitudinal data consists of six sessions of 140 trials of EEG data, with a total number of 840 trials. P2′s longitudinal data consists of three sessions of 140 trials of EEG data, with a total number of 420 trials.

A 32-channel g.Nautilus EEG acquisition system was used to record the EEG data with a sampling rate of 500 Hz. Figure 2 depicts the EEG acquisition montage. During EEG data recording, the participant responded using a mouse-like button using the stroke impaired upper limb. The detailed training protocol and experiment setup were the same as described in [10]. For each data acquisition session, EEG data were acquired from at least 40 responses. All datasets for this study are available from the corresponding author upon request.

### 2.4. Data Pre-Processing

In accordance with the data pre-processing and model generation process described in [10], the channel order of recorded EEG data was initially processed based on the side that the participant used in the EEG data acquisition. For example, if the participant responded using their left hand during the data acquisition, the EEG channels were flipped to match those of right-hand responses in order to integrate the data.

Within the two-minute recording, raw EEG data were filtered using a 1–45 bandpass Finite Impulse Response (FIR) filter. Event responses that were less than 9 s apart were discarded to accommodate for the 8–10 s, event-related synchronization and desynchronization (ERS/ERD) process [4,5,17,18,19,20]. The data were then truncated into 6 s trials according to the events. Five trials were extracted from each event, starting from 4 s before the response event to 3 s before the event, with a step size of 200 ms and fixed window size of 6 s. This multiple sample extraction method has been proven to be effective in neural network-related research to expand the sample size [21,22,23]. Fast Fourier Transformation (FFT) was used to calculate the band power and phase response of the EEG data in each trial for later processing. As the original EEG data were filtered with 1–45 Hz FIR filter, the frequency and phase components associated with frequency <1 Hz or >45 Hz were discarded after FFT. With 500 Hz sampling rate, 6 s as the signal length, the frequency band between 1 and 45 Hz resulted in 270 elements in power spectrum density (PSD) and phase information. Using all the 32 channels of the EEG acquisition system, stacking the power spectrum density and the phase information, each trial of the EEG data was in the shape of a 270 × 32 × 2 matrix, after pre-processing.

### 2.5. Neural Network Model Configuration, Hyperparameter Optimization and Testing

Open source Python toolboxes such as Keras [24] and tensorflow [25] were used to configurate the ANN methods used. Apache Spark [26] was introduced to facilitate parallel training of the proposed ANN on computer clusters maintained by Compute Canada. Specific graphic process unit (GPU) nodes of Compute Canada was used in this study, with two Intel E5-2683 v4 Broadwell CPUs (16 cores) and two NVIDIA P100 Pascal GPUs on each node. The proposed ANN models were trained based on the FMA score of the more impaired upper extremity.

The first ANN configuration was a CNN, similar to the configuration introduced by Zhang et al., in [10]. Table 1 depicts the CNN model configuration and hyperparameters. The hyperparameters were also optimized jointly as suggested in [10] using Tree Parzen Estimator (TPE) [27]. The training and hyperparameter tuning process of CNN took 4 h of four GPU nodes, which contain 128 CPU cores and 8 GPUs.

ResNet was also trained to investigate the feasibility of improving the accuracy in assessing motor function with deeper neural networks. Therefore, the hyperparameter space of the ResNet model was mostly focused on the depth of the model configuration. Different from deep learning for image recognition application, the number of features and the dimension of the features were not fully investigated for EEG applications in the literature, especially for motor assessment applications. As a result, the number of filters in the first ResNet stage and size of the kernels in the second layer in each ResNet block were also tuned to improve the model performance. Based on the ResNet model configuration in the original paper, the hyperparameters on the other layers of ResNet were determined by matching the data dimensionality accordingly [14]. Figure 3 depicts the ResNet configuration and hyperparameters used in this paper. The proposed ResNet used three stages, three fully connected layers, and one dropout layer, as shown in Figure 3a. In each ResNet stage, multiple ResNet blocks were configured as shown in Figure 3b. The hyperparameters were also tuned jointly using TPE. For ResNet, the model generation and hyperparameter optimization process took 12 h of ten GPU nodes, which contain 320 CPU cores, and 20 GPUs.

The two proposed ANN models generated from the training data were tested with the EEG data collected from P1 and P2 during the baseline assessment and training. Given that P1 and P2 also participated in the pool data collection, the performance of the proposed ANN models should be evaluated without P1 and P2′ data involved in the training. This was done in order to exclude the prior bias on the model generation. However, due to the complexity of the computation required and limitation of available resources, only P1 was evaluated in a cross-participant paradigm.

### 2.6. Statistical Analysis

FMA is a long-established motor assessment tool which has been widely used in the stroke related research studies [28]. In this study, P1 and P2 were involved in the longitudinal data acquisition. The two one-sided test (TOST) was used to evaluate whether the distribution of the motor function assessment from the proposed two ANN models was equivalent to the distribution of the FMA score (*p* < 0.05). According to the literature, for 66 scores of upper-extremity subsection in FMA, Van der Lee et al. found the test-retest agreement was −5 to 6.6 points for the FMA upper extremity sub-section [29]. Sanford et al. discovered that the measurement error was ±7.2 points for the FMA upper extremity sub-section [30]. Hsueh et al. reported the smallest real difference of the FMA was 10% of the highest score (i.e., ± 6.6 points in FMA for upper extremity) [31]. Hiragami et al. suggested the minimal clinically important difference of the FMA was a score of 12.4 for upper extremity [32]. In order to improve the confidence of the statistical analysis, the smallest equivalence interval of TOST was used in the statistical analysis (−5 to 6.6 points), which is about 10% of the full score of FMA assessment for upper extremity.

## 3. Results

### 3.1. Demographic Data

Two participants with stroke were recruited for this study; both survived a stroke seven years ago. Table 2 provides the detailed demographic data of the two participants. Both of the participants were right-handed before stroke, and P2 switched handedness due to the impairment of stroke. P2 also experienced minor cognitive impairment, with MoCA scores still within the inclusion criteria [16].

### 3.2. ANN Model Training and Hyperparameter Optimization

The proposed ANN model configurations were trained with the pool data, and the hyperparameters were also optimized with TPE. In the pool data, 140 trials of EEG data were extracted for each participant, which were used in the training. Thirty (30) trials of EEG data, which were not involved in the training, were used as a test set to evaluate the obtained model. As the performance of the CNN method has been extensively discussed in [10], only ResNet method was evaluated in this section. Figure 4 shows the ResNet result. Healthy participants’ assessment results were not included as they clustered in a small area, in order to avoid biasing the correlation analysis. The correlation analysis suggested equivalent prediction accuracy with the proposed ResNet method (r = 0.986) as reported in [10].

### 3.3. Longitudinal Within-Participant Test Result and Prediction

As described in the Methods section, the trained ANN models evaluated P1 and P2 EEG data collected in a longitudinal fourteen-week program. P1 was able to complete the whole fourteen-week program with six assessment sessions. P2 dropped out from the program after four weeks of baseline assessments. Motor function scores generated from the ANN models were obtained based on a single trial of EEG data. The estimated assessment scores were averaged with all the trials collected on the same day of the data acquisition. For the WMFT, the time to finish each item of the WMFT was recorded. An average value was reported for each WMFT assessment session, indicating the participant’s average ability to finish one item in the test. Figure 5 summarizes the test results. On average, the ResNet model was able to assess P1′s FMA score of upper-extremity. The ResNet model was able to assess P1′s FMA score with a mean absolute error (MAE) of 1.031 points; TOST suggests that the assessment scores and the FMA scores were from the same distribution (*p* = 2.047 × 10^−4^). The CNN model was able to assess P1′s FMA score with MAE of 1.217 points; TOST suggests that the CNN assessment scores and the actual FMA scores were from the same distribution (*p* = 4.818 × 10^−4^).

Figure 6 shows the longitudinal test results for P2. The ResNet model was able to assess FMA with MAE of 3.409 points, and CNN model was able to assess FMA with MAE of 1.965 points. Due to the fact that P2 only participated in three session, the power of the statistical analysis is low. For the ResNet method, the assessment scores from EEG were not statistically equivalent to the FMA score (*p* = 0.231), yet the two scores were not statistically different (*t*-test, *p* = 0.281). For the CNN model, the assessment scores from the EEG were not equivalent to the FMA score (*p* = 0.0781), yet the two groups of scores were not statistically different (*t*-test, *p* = 0.553).

In the longitudinal within-participant test of P1 and P2, the ResNet model was able to assess upper extremity FMA with MAE of 2.22 points. The CNN model had a superior performance, which was able to assess upper extremity FMA with MAE of 1.59 points.

### 3.4. Longitudinal Cross-Participant Test Result and Prediction

Given that P1 and P2 were also participants who participated in the pool data collection, the two proposed ANN models were also evaluated in a cross-participant paradigm. Only P1 was evaluated in this section due to the high computation resource requirement of the proposed ANN models. Figure 7 shows the longitudinal cross-participant test results. The ResNet model was able to assess motor function of the upper extremity FMA with MAE of 4.085 points, which is statistically equivalent to the FMA score (*p* = 0.0326). For the CNN model, the results showed that the MAE between the assessment from EEG and FMA was 1.302 points. The motor function assessment scores from the CNN model and the FMA scores were also statistically equivalent according to TOST (*p* = 1.860 × 10^−3^).

The cross-participant models were also evaluated with the three-session baseline data from P2. Since P2′s EEG data were also in the pool data, the test results presented in this section were not technically cross-participant test results. The ResNet model was able to estimate upper extremity FMA with MAE of 2.61 points. The ResNet model showed higher accuracy compared with CNN model accuracy (with MAE 3.74 points) in P2 longitudinal cross-participant testing.

Summarizing the longitudinal assessment accuracy with P1 and P2, the ResNet model was able to assess upper extremity FMA with MAE of 3.90 points, and showed superior performance compared to the CNN model, whose MAE was 4.01 points.

## 4. Discussion

In this study, a CNN model and a ResNet model were configured and trained using pool data collected in previous study, which consisted of EEG data from 12 healthy participants and 14 participants with chronic stroke. Then the obtained two ANN models were tested with two participants’ EEG data collected in a longitudinal paradigm. P1 received three baseline assessments and 18 sessions of training. P2 completed only three baseline assessments. Motor functions were assessed during the baseline, training, and retention sessions. On average, P1 scored 45.3 of 66 points in FMA upper extremity section. P2 scored 52.0 of 66 points. Both ANN models were able to assess the FMA result using EEG data with MAEs smaller than the variation range of FMA [31]. The average MAEs were 2.220 points for the ResNet model and 1.591 points for the CNN model in the within-participant test results. P1 and P2 also participated in the pool data collection about 12 months before the longitudinal data collection. In order to eliminate bias introduced by prior data in the model generation, the ResNet model and the CNN model were also tested in a cross-participant paradigm, using models trained without P1′s EEG data. The results suggested both ANN models were still able to assess P1′s upper extremity FMA scores in the cross-participant test, with slightly increased MAE (2.294 points for ResNet model, 1.302 points for CNN model). Both ANN models were able to estimate upper extremity motor function with an MAE smaller than FMA accuracy, which is about 10% of the full score, according to the psychometric property analysis of FMA [29,30,31,32]. Therefore, the ResNet model and the CNN model may have higher longitudinal accuracy in assessing upper extremity motor function, compared to FMA.

The sample size was generally small and almost half of the data were from healthy participants. In [10], the effect of the sample size and uneven distribution of the participants‘ status was extensively discussed in the CNN model training. The cross-participant (i.e., leave-one-out cross validation) test results and the model training with incremental data from healthy participants (Figure A1 in [10]) strongly supports the conclusion that the data from 12 healthy participants improve the CNN model’s performance. However, we were unable to repeat the same process in this paper with ResNet configuration due to the large computational resource requirement of ResNet. Instead, we excluded P1′s data in the pool data and tested the cross-participant performance with P1′s EEG data collected during the longitudinal training (Section 3.4). The results showed comparative MAE between ResNet and CNN model, which suggested the sample size of the pool data is sufficient for the concept validation study presented in this paper.

The ResNet model was included in this study to investigate the necessity of a complex and “deep” model configuration in motor function assessment. The ResNet model did show smaller MAE in the within-participant test for P1 compared to the CNN model: the difference is not significant (paired t-test, *p* = 0.722). However, the CNN model showed slightly higher accuracy in the within-participant test in P2 and the following cross-participant test for P1. The upper extremity motor function distribution of the pool data and the specific model configuration might both contribute to the performance inconsistency. For example, the upper extremity FMA scores of the pool data participants with chronic stroke naturally formed three major clusters: 10–24 points, 36–41 points, and 45–51 points. P1 and P2′s average upper extremity FMA scores were 45.3 points and 52.0 points, which happened to be at the beginning and end of the last cluster interval. Removing P1′s data in the cross-participant test might have significantly altered the curvature of the regression process, due to missing key data in the regression process. These results suggested that the CNN model was more sensitive to data amount changes; even the ResNet model had significantly more trainable parameters compared to the CNN model. These results might be due to the special identity mapping configuration of the ResNet model [14]. Motor function assessment scores from the two ANN models were statistically equivalent, both in the within-participant test (TOST, *p* = 2.45 × 10^−4^; paired *t*-test, *p* = 0.494) and in the cross-participant test (TOST, *p* = 0.0119; paired *t*-test, *p* = 0.143). These findings suggest the ResNet model might have higher accuracy with more data involved in the model generation.

The averaged WMFT results were negatively correlated with the FMA data for both participants in this study. Some sessions had a WMFT score increase compared to the score recorded in the previous session, for unknown reasons, which suggested motor function deterioration in such sessions. The Assessment 1 session of P1 and Baseline 2 of P2 are the two typical sessions in cross-participant test. It is interesting to see that both two ANN models had bad FMA score estimation accuracy on these two sessions. Considering the fact that the cross-participant test removed the participants’ prior data in the training set, the ANN models were not able to overfit to learn the participants’ identity from the EEG data. Therefore, this is another piece of evidence suggesting that the two ANN models might have higher accuracy and robustness in assessing motor function, compared to FMA.

In this study, two participants with chronic stroke were invited to participate in longitudinal EEG acquisition. While intended as a preliminary validation, the major limitation of this study was the sample size. Although the results are promising, further replication studies with larger patient samples are required to support the generalizability of the results. In addition, due to the intrinsic FMA score distribution of the training data, the 14 participants’ motor function performance naturally formed three clusters. P1 and P2 participants were located in the same FMA score cluster interval (45–51 points). Therefore, the longitudinal performance of the two ANN methods were not evaluated comprehensively with the full range of the motor function assessment. In the future, a longitudinal study on participants with chronic stroke of various motor function statuses should be performed to fully investigate the longitudinal efficacy of the proposed ANN models.

## 5. Conclusions

The current study evaluated the feasibility of using EEG to score upper extremity motor function using a longitudinal paradigm with two participants involved in stroke rehabilitation. The results suggested the motor function scores from ResNet and CNN models were correlated with upper-extremity FMA scores using a longitudinal design, with both yielding good accuracy and reliability, supporting the feasibility of using EEG as an objective physiological measure for motor function recovery.

## Figures and Tables

**Figure 1 sensors-20-05487-f001:**
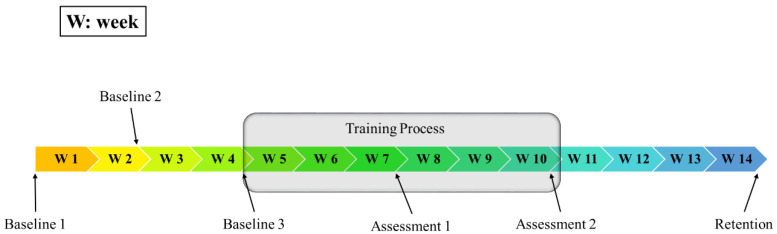
Timeline arrangement for the fourteen-week program.

**Figure 2 sensors-20-05487-f002:**
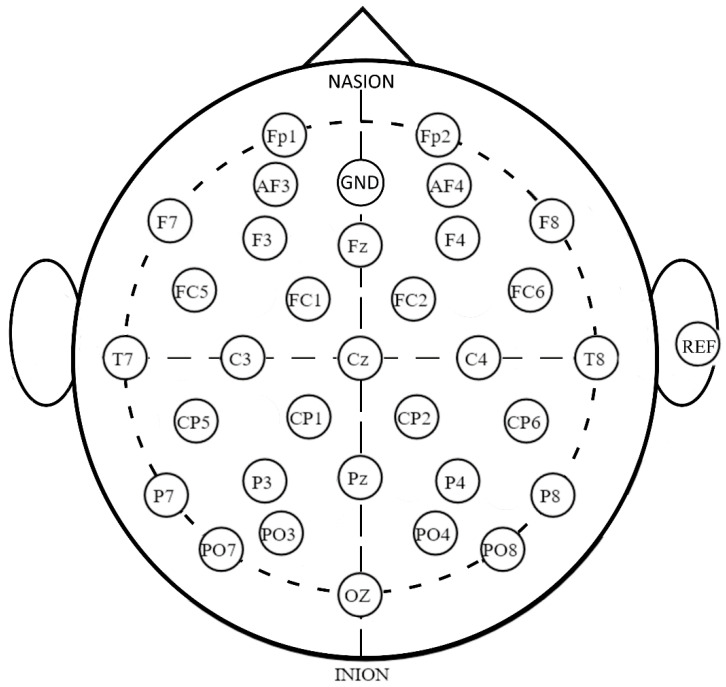
The montage for g.Nautilus electroencephalography (EEG) 32 channel acquisition system, GND is the ground channel, REF is the reference channel.

**Figure 3 sensors-20-05487-f003:**
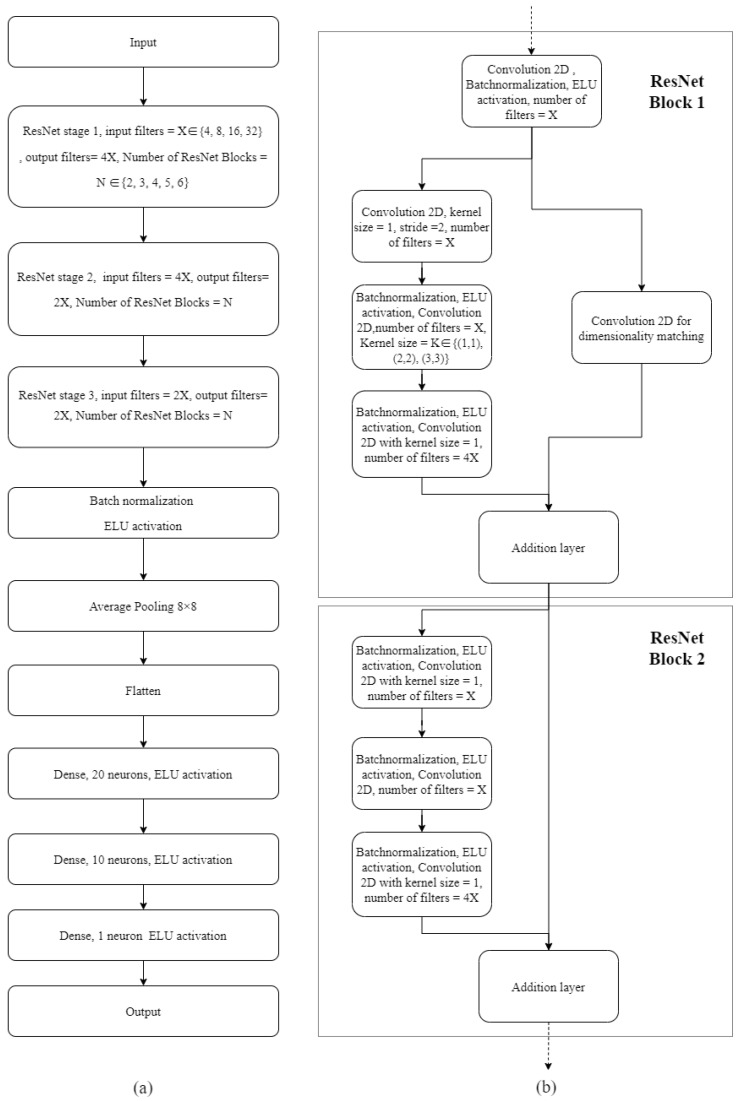
Residual neural network (ResNet) structural configuration. (**a**) is the overall residual neural network configuration which consists of three ResNet stages, fully connected layers, and dropout layers. The ResNet stage consists of multiple ResNet blocks; (**b**) is the network configuration for the ResNet stage 1, which consists of a certain number of ResNet blocks; stage 2 and stage 3 consist of multiple blocks as shown in ResNet block 2. Batch size ∈ {16, 32, 64} and training iterations ∈ {100, 150, 200} were also included in the hyperparameter optimization.

**Figure 4 sensors-20-05487-f004:**
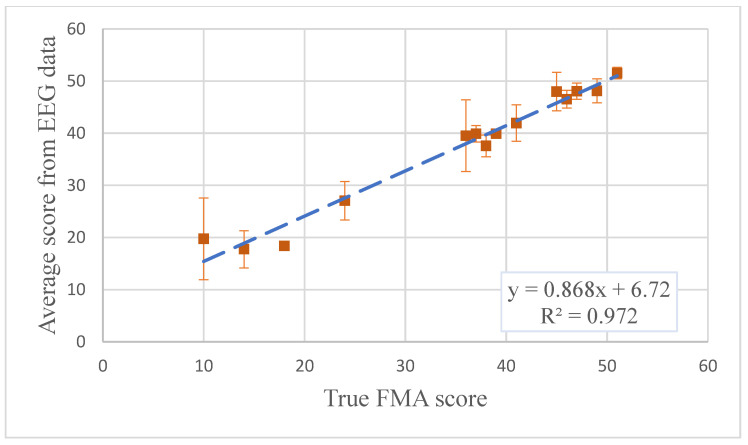
The model test result for ResNet method (*n* = 14); the healthy participants’ results were not included in the correlation analysis as their data were clustered around the (66, 66) area, which would affect the correlation analysis results. The assessment scores were produced with one trial of EEG data and averaged across all trials from one participant.

**Figure 5 sensors-20-05487-f005:**
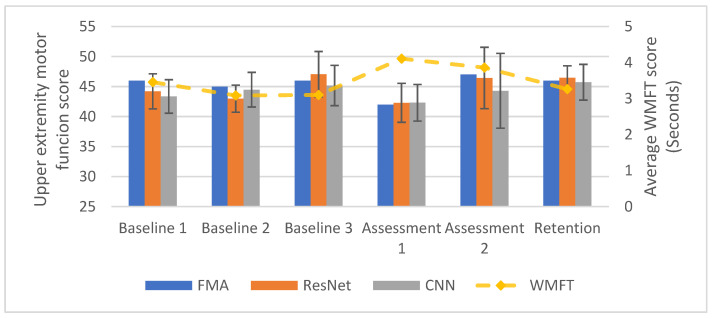
The longitudinal test results of the ANN models with EEG data collected from P1. Assessment scores from EEG data were produced based on single trials of EEG data and averaged across all EEG trials collected on the same day.

**Figure 6 sensors-20-05487-f006:**
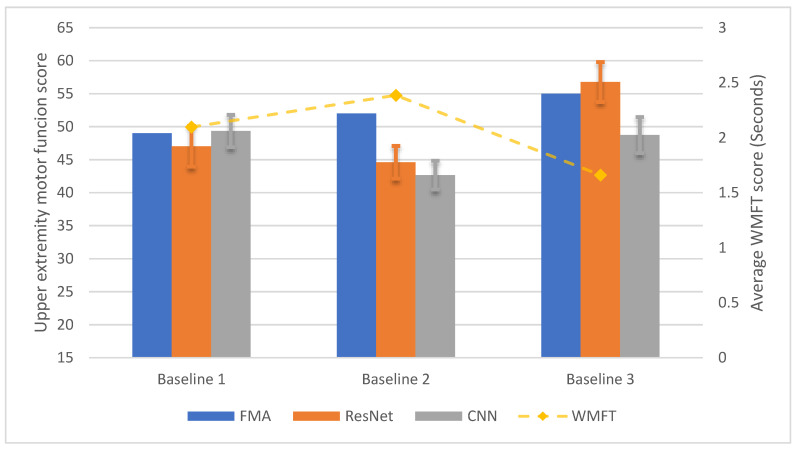
The longitudinal test results of the ANN models with EEG data collected from P2.

**Figure 7 sensors-20-05487-f007:**
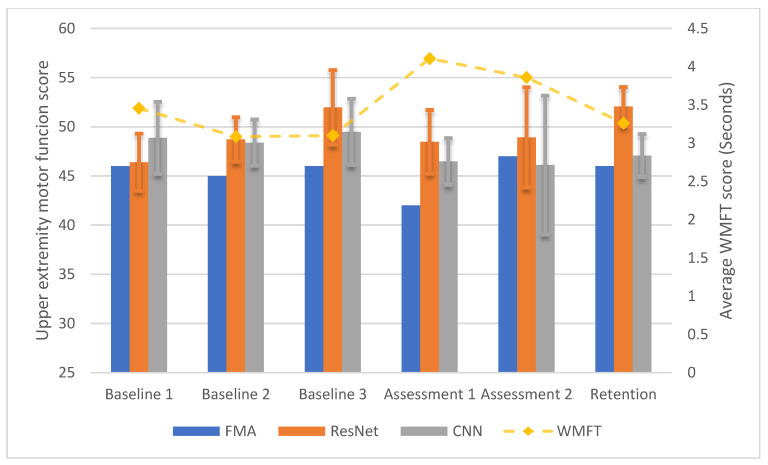
The longitudinal test results of the ANN models with EEG data collected from P1, when P1′s data were removed from the pool data, which was collected approximately 12 months before the study introduced in this paper.

**Table 1 sensors-20-05487-t001:** Convolutional neural network (CNN) structural configuration. All the convolution layers and fully connected layers (dense layers) used exponential linear unit (ELU) as activation function. Batch size ∈ {64, 128} and training iterations ∈ {100, 150, 200} were also included in the hyperparameter optimization.

Layer Name	Parameters
Input Layer	Shape = (None, 270, 32, 2)
2D Convolutional Layer 1	Number of Filters ∈ {25, 50, 100}, Kernel Size ∈ {(2,2), (2,4), (2,6), … (10,8), (10,10)}
Max Pooling Layer 1	Pooling Size = (2, 2)
2D Convolutional Layer 2	Number of Filters ∈ {25, 50, 100}, Kernel Size = (2, 2)
Max Pooling Layer 2	Pooling Size = (2, 2)
Flatten Layer	N/A
Dense Layer 1	Number of neurons = 100
Dropout Layer	Dropout rate∈ {0 to 1 with 0.05 as step size}
Dense Layer 2	Number of neurons = 25
Dense Layer 3	Number of neurons = 10
Dense Layer 4	Number of neurons = 5
Output Layer	Number of neurons = 1

**Table 2 sensors-20-05487-t002:** Demographic Data.

ID	Age	Gender	Years after Stroke	Affected Side	MOCA	Handedness before Stroke
P1	65	Male	7	Left	27	Right
P2	51	Female	7	Right	23	Right

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
