# Peer review of "The Feasibility of Longitudinal Upper Extremity Motor Function Assessment Using EEG"

_sensors, 2020, doi:10.3390/s20195487_

Round 1
Reviewer 1 Report
Interesting work for identifying stroke patients.
The authors have used 14 healthy subjects and 12 patients with stroke. The ratio is not correct as the stroke patient cases need to be more than the control cases. The dataset is too small to use for this type of training, in particular, deep learning requires much more data to learn from. There is an indication to use previous dataset, but it not clear when it was used and how many cases are involved.
The results show comparison between WMFT which is considered s the gold standard and proposed techniques (CNN, ResNet). The proposed techniques are similar to FMA, while WMFT show difference, in particular for assessments 1 and 2.
Some part of the paper is written in the past tense, "The aim of this paper was ..", which is not the correct way, please use present tense.
Author Response
The authors have used 14 healthy subjects and 12 patients with stroke. The ratio is not correct as the stroke patient cases need to be more than the control cases. The dataset is too small to use for this type of training, in particular, deep learning requires much more data to learn from. There is an indication to use previous dataset, but it not clear when it was used and how many cases are involved.
We thank the reviewer for the comments, please see our replies below.
Sorry for the mis-understanding in the introduction section, the two ANN models were generated with 3640 trials EEG data from 12 healthy participants and 14 participants with chronic stroke (referred to as “pool data” in this manuscript, previous dataset from [5]). The generated models were tested with the “longitudinal data” from P1 and P2, which were recorded in 14 weeks and 4 weeks respectively. We have updated the manuscript (attached below) to clarify this issue, please see Section 2.3 Paragraph 1.
We agree with that sample size is always a concern that may affect the result of the ANN model performance. This often must be balanced against challenges related to stroke patient related research. We recruited as many patients as possible, and were then able to be guided from prior research for smaller sample size analyses. In our previous paper [5], we explored sample size and uneven distribution issues using the CNN model configuration, confirming model performance without over-fitting the model on training set. We have updated the Discussion section Paragraph 4 in the manuscript to clarify this point.
The results show comparison between WMFT which is considered as the gold standard and proposed techniques (CNN, ResNet). The proposed techniques are similar to FMA, while WMFT show difference, in particular for assessments 1 and 2.
Some part of the paper is written in the past tense, "The aim of this paper was ..", which is not the correct way, please use present tense.
We have updated the manuscript about this issue, please see line 55 in the updated manuscript.

Reviewer 2 Report
Authors of this manuscript aims to help evaluate stroke patient recovery based on motor function, while finding that EEG can be one potential tool for such task. To better understand such possibility, authors propose for a longitudinal paradigm evaluation, where experiments provide positive preliminary results for such application. The proposed processes in the manuscript is explicitly presented with details, and with considerably detailed result presentation and discussion. At a glance I cannot tell how this is a longitudinal study... In section 2.3 it mentions data collection in [5] and in 3.2 and describes the tested patients and control group subjects, while it is not quite clear how your data is collected and what is your data looks like. More insights on the utilised two neural networks are expected. Such as how generally it is constructed in your case, is it different to their original proposal, or why the hyperparameters are set as it is in your work? Better organization of content is recommended, especially for very large figures. If scale is not feasible, please put then into appendix with references. Also tables with hyperparameters can be arguably merged with the figures of network structures.
Author Response
Authors of this manuscript aims to help evaluate stroke patient recovery based on motor function, while finding that EEG can be one potential tool for such task. To better understand such possibility, authors propose for a longitudinal paradigm evaluation, where experiments provide positive preliminary results for such application.
The proposed processes in the manuscript is explicitly presented with details, and with considerably detailed result presentation and discussion. At a glance I cannot tell how this is a longitudinal study... In section 2.3 it mentions data collection in [5] and in 3.2 and describes the tested patients and control group subjects, while it is not quite clear how your data is collected and what is your data looks like.
The authors thank the comments from the reviewer, please see our replies below. We have updated Section 2.3, Paragraph 1 to clarify the data sets used in this study, the updated manuscript was attached below.
We used two data sets to investigate the longitudinal performance of the ANN models, the pool data and the longitudinal data. The pool data, from [5], were used to train the ANN models. The longitudinal data were collected from two participants with chronic stroke (P1 and P2 in the manuscript) in 14 weeks and 4 weeks respectively, which were used to test the obtained ANN models in order to evaluate the longitudinal performance of the proposed motor function assessment method based on EEG. P1 and P2 also participated pool data collection about 12 months before the longitudinal data collection.
More insights on the utilized two neural networks are expected. Such as how generally it is constructed in your case, is it different to their original proposal, or why the hyperparameters are set as it is in your work? Better organization of content is recommended, especially for very large figures. If scale is not feasible, please put then into appendix with references. Also tables with hyperparameters can be arguably merged with the figures of network structures.
Great question. Please see Section 2.5, Paragraph 3 in the updated manuscript. Also, based on the reviewer’s suggestion, Figure 3 and Figure 4 has been updated to include the hyperparameters in the model hyperparameter optimization, the hyperparameter tables have been removed from the manuscript.
The idea of hyperparameter optimization was to fine tune the network to achieve the best performance. For the CNN model, the hyperparameter space in this paper was pretty much the same with reference [5], which was proven to have good performance in cross-participant test. For the ResNet model, the idea was to test the possibility of improving the model performance with deep neural network configuration. Therefore, the hyperparameter space of the ResNet model was mostly focused on the depth of the model configuration. Different from deep learning for image recognition, the number of features and the dimension of the feature were not fully investigated for EEG applications in the literature, especially for motor assessment applications. As a result, the number of filters in the first ResNet stage and size of the kernels in the second layer in each ResNet block were also tuned to improve the model performance. Based on the ResNet model configuration in [7], the hyperparameters on the other layers of ResNet was determined via matching the data dimensionality accordingly.

Round 2
Reviewer 2 Report
Most issued are resolved in this submission, yet minor revision is recommended, and this time concerns mostly are on:
1) as preliminary work, the verification of the feasibility of proposed work is acceptable while still investigation from more subjects are recommended; Study of 2 subjects can be demonstrative, yet hardly decisive as proof;
2)The figure 3 and 4 generally is clear, while reconstruction showing the actual neural networks are expected, otherwise they can be presented as tables for concision.
3) It is still not quite distinguishable from your work to existing literature. Contributions are acknowledged, yet limitedly.
Author Response
We thank reviewers for the comments, please see the attachment for the replies.
